Diagnosis and treatment of epididymal tuberculosis: a review of 47 cases

Man Jiangwei
Cao Lei
Dong Zhilong
Tian Junqiang
Wang Zhiping
Yang Li ery_yangli@lzu.edu.cn
Department of Urology, Lanzhou University Second Hospital , Lan Zhou , China
Tulkens Paul
Electronic publication date: 2020 Jan 6
Publication date: 2020
Volume: 8
Electronic Location ID: e8291
Received 2019 Aug 7; Accepted 2019 Nov 24
Copyright: © 2020 Man et al.
Copyright year: 2020
Copyright holder: Man et al.
License: This is an open access article distributed under the terms of the Creative Commons Attribution License, which permits unrestricted use, distribution, reproduction and adaptation in any medium and for any purpose provided that it is properly attributed. For attribution, the original author(s), title, publication source (PeerJ) and either DOI or URL of the article must be cited.
License URL: https://creativecommons.org/licenses/by/4.0/

Keywords: Epididymal tuberculosis, Clinical characteristics, Chemotherapy

Funding: Cuiying Scientific and Technology Innovation Program of Lanzhou University Second Hospital CY2018-BJ02 This paper is supported by Cuiying Scientific and Technology Innovation Program of Lanzhou University Second Hospital (No. CY2018-BJ02). There was no additional external funding received for this study. The funders had no role in study design, data collection and analysis, decision to publish, or preparation of the manuscript.

==============================
Objective

To analyze the clinical manifestations, diagnosis and treatment outcomes in a series of patients with epididymal tuberculosis.

Methods

This study is a retrospective data analysis of 47 cases of histologically-confirmed epididymal tuberculosis in patients treated at our hospital from November 2012 to December 2018.

Results

The average age of the patients was approximately 42 years. The epididymal lesion location was left-sided in 15 patients (31.9%), right-sided in 22 patients (46.8%) and bilateral in 10 patients (21.3%). The main symptoms were painless swelling of the scrotum in 21 cases (44.7%) and scrotal drop pain in 21 cases (44.7%). Scrotal physical examination revealed epididymal beaded enlargement in 12 patients (25.5%), testicular mass in one patient (2.1%), scrotal tenderness alone in seven patients (14.9%), ill-defined epididymal-testicular border in 21 patients (44.7%) and sinus formation in six patients (12.8%). After 2–4 weeks of anti-tuberculosis chemotherapy, the patients underwent a surgical procedure. We found that 10 (83.3%) of the 12 patients whose main symptom was epididymal beaded enlargement underwent simple epididymal surgery. Of the 21 patients whose main clinical manifestation was ill-defined testis-epididymis demarcation, 16 (72.2%) underwent epididymis-testicular surgery. All patients underwent postoperative chemotherapy for 3–6 months. Postoperative follow-up showed good response to treatment.

Conclusion

It is difficult to diagnose early-stage epididymal tuberculosis. Epididymal tuberculosis is likely to have invaded surrounding tissues when signs such as epididymal beaded changes and ill-defined epididymis-testis border are present. Surgical treatment combined with preoperative and postoperative chemotherapy is an effective approach to treating this condition.

Introduction

Epididymal tuberculosis is a rare extrapulmonary form of tuberculosis that occurs in young adults (Surati, Suthar & Shah, 2012). Patients with this disease may have no obvious clinical symptoms or only mild symptoms. The disease typically develops slowly and early diagnosis is difficult; delayed diagnosis and misdiagnosis are common. Recently, due to the emergence of multi-drug resistant bacteria, anti-tuberculosis drug resistance, and the widespread use of glucocorticoids, the incidence of male genital tuberculosis, including epididymal tuberculosis, has been increasing worldwide. Epididymal tuberculosis remains an important health problem in many developing countries, including China (Kulchavenya et al., 2012). The pathological features of epididymal tuberculosis are extensive tissue destruction and fibrosis, eventually leading to destruction of the epididymis and surrounding genital tissues and organs and complications such as infertility and other serious effects on male reproductive system function.

At present, early diagnosis and regular chemotherapy are the keys to cure male reproductive system tuberculosis and to avoid surgical treatment. However, due to its late onset and nonspecific clinical signs and symptoms, and the lack of rapid, sensitive, and specific diagnostic methods, the disease is often misdiagnosed or the diagnosis is delayed. Therefore, surgical treatment combined with chemotherapy has been the preferred treatment approach to this disease. This study, a retrospective analysis of a series of cases of epididymal tuberculosis, is intended to more fully characterize the clinical features and outcomes of surgical treatment of this disease.

Material and Methods

We collected the medical records of 47 patients with histologically-diagnosed epididymal tuberculosis from November 2012 to October 2018 at the Second Hospital of Lanzhou University. Age, clinical signs and symptoms, diagnostic methods, and treatment effects were collected from the patients’ electronic medical records, and the final results were compared and analyzed. The following information was supplied relating to ethical approvals (i.e., approving body and any reference numbers): Ethics Committee of Lanzhou University Second Hospital Number: 2019A-079. Written consent was obtained from all patients.

Results

The mean age of the patients was 41.98 years (range, 19–72 years); 15 patients (31.9%) had tuberculosis in the left epididymis, 22 patients (46.8%) had tuberculosis in the right epididymis and 10 patients (21.3%) had bilateral epididymal tuberculosis. Of the 47 patients, eight (17.0%) had a history of tuberculosis, including three cases of pulmonary tuberculosis, four cases of renal tuberculosis and one case of prostate tuberculosis (Table 1).

Table 1 The clinical manifestations and diagnosis in 47 patients with epididymal Tuberculosis.

	Number	Percentage (%)	
Location	
Left	15	31.90	
Right	22	46.80	
Bilateral	10	21.30	
Symptoms	
Painless swelling	21	44.70	
Scrotal drop pain	21	44.70	
Urinary tract irritation	4	8.50	
Scrotal skin ulceration	1	2.10	
Systemic symptoms	
Yes	6	12.80	
No	41	84.20	
Physical examination	
Epididymal beaded enlargement	12	25.50	
Testicular mass	1	2.10	
Scrotal tenderness	7	14.90	
Ill-defined epididymal testicular border	21	44.70	
Sinus formation	6	12.80	
History of tuberculosis	
Pulmonary tuberculosis	3	0.06	
Renal tuberculosis	4	0.08	
Prostate tuberculosis	1	0.02	
No	39	0.84	
Preoperative diagnosis	
Tuberculosis	41	87.20	
Tumors	3	6.40	
Masses	3	6.40	

Main symptoms at the time of onset included: painless swelling of the scrotum in 21 patients (44.7%); scrotal drop pain in 21 patients (44.7%); urinary tract irritation such as urinary frequency, dysuria and hematuria in four patients (8.5%); and scrotal skin ulceration in one patient (2.1%). Systemic symptoms such as low-grade fever, fatigue and night sweats occurred in six patients (12.8%). All patients underwent scrotal physical examination, which revealed epididymal beaded enlargement in 12 patients (25.5%), testicular mass in one patient (2.1%), scrotal tenderness alone in seven patients (14.9%), ill-defined epididymal testicular border in 21 patients (44.7%) and sinus formation in six patients (12.8%) (Table 1).

Urinalysis and chest imaging were performed in all patients. Twenty patients (42.6%) were positive for white blood cells in urine and eight patients (17.0%) were positive for red blood cells in urine. Chest imaging was positive in eight patients (17.0%). The preoperative diagnoses were epididymal tuberculosis in 41 patients (87.2%), epididymal tumor in three patients (6.4%) and epididymal mass in three patients (6.4%). All patients had evidence of surgical treatment of the lesion.

Among the 47 cases, 41 (87.2%) were initially identified as epididymal tuberculosis and treated pre-operatively with rifampicin, isoniazid, pyrazinamide, and ethambutol for 2–4 weeks. Twenty-six patients (55.3%) had epididymectomy and 21 patients (44.7%) had epididymectomy combined with orchiectomy, and all recovered well after surgery.

We found that 10 (83.3%) of the 12 patients whose main symptom was epididymal beaded enlargement underwent simple epididymal surgery and two (16.7%) underwent epididymis-testicular surgery. Of the 21 patients whose main clinical manifestation was ill-defined testis-epididymis demarcation, 16 (72.2%) underwent epididymis-testicular surgery and five epididymal surgery (Table 2).

Table 2 Association between surgical methods and physical examination.

	Total	Epididymal resection	Epididymis-testicular resection	
Ill-defined epididymal testicular border	21	16 (76.2%)	5 (23.8%)	
Epididymal beaded enlargement	12	10 (83.3%)	2 (16.7%)	
Scrotal tenderness	7	7 (100%)	0	
Sinus formation	6	3 (50%)	3 (50%)	
Testicular mass	1	1 (100%)	0	

Macroscopically, all specimens contained solitary or confluent pale, grayish caseous necrotic nodules. Some lesions invaded the entire epididymis, some showed involvement of the testis with adherence to the scrotum to form a cold abscess and some showed the formation of ulcerated sinus tracts in the skin. Microscopically, the centers of the specimens were red-stained with amorphous granular, irregularly sized foci of caseous necrosis surrounded by tuberculous granulation tissue (epithelioid cells, Langhans giant cells and lymphocytes) (Figs. 1 and 2).

Figure 1 Microscopy images of caseous necrosis with amorphous granular.

Figure 2 Microscopy images of irregularly sized foci of caseous necrosis surrounded.

All patients received anti-tuberculosis treatment with rifampicin, isoniazid, pyrazinamide, or ethambutol after surgery for 3–6 months. No recurrence was reported.

Discussion

The number of tuberculosis patients in China accounted for 10–12% of the total number of tuberculosis patients in the world, ranking second in the world (Glaziou, Floyd & Raviglione, 2018; Zhang et al., 2016). People in Gansu Province of China have a high incidence of tuberculosis and are relatively delayed in seeking medical care (Zhang et al., 2016). The presentation of tuberculosis is increasingly atypical. Unfortunately, the prevalence of drug-resistant strains of tuberculosis is increasing (Lee et al., 2015). The main treatment of epididymal tuberculosis is early anti-tuberculosis treatment. However, since most of the patients with epididymal tuberculosis in Gansu are in the terminal stage, surgical treatment combined with chemotherapy has been the best treatment modality for this type of patients.

Reproductive system tuberculosis can occur in any age, mainly in men 30–50 years old. Due to the long incubation period, it is not common in children. The most commonly involved organ is the epididymis, followed by the seminal vesicle, prostate, testis, and vas deferens (Yadav et al., 2017). Isolated epididymal tuberculosis is very rare (Gueye et al., 1998). One new study indicated that isolated epididymal tuberculosis may be the first or only manifestation of early genitourinary tuberculosis (Viswaroop, Kekre & Gopalakrishnan, 2005). Similarly, in our patient cohort, there were 39 cases (82.9%) of isolated epididymal tuberculosis.

The pathogenesis of epididymal tuberculosis includes blood-borne transmission and transurethral reflux of Mycobacterium tuberculosis caused by factors such as trauma, alcohol abuse and excessive sexual activity (Tzvetkov & Tzvetkova, 2006). Epididymal tuberculosis lesions first appear in the tail of the epididymis, owing to its rich blood supply and to retrograde infection from the vas deferens. The lesions of epididymal tuberculosis gradually invade the body to the head, finally affecting the entire epididymis. In severe cases, the testis can be involved (Chung et al., 1997). In this group, 21 patients (44.7%) with epididymal tuberculosis invading the testis underwent radical surgery of the testicle. In our study, the main cause of the patient’s visit was painless mass of the epididymis, which is consistent with previous studies (Kho & Chan, 2012). However, the proportion of patient with scrotal pain is higher, mainly because the patients included in this study all had advanced epididymal tuberculosis or involvement of the testis or scrotum.

The gold standard for diagnosing tuberculosis is the isolation and culture of M. tuberculosis. In cases of suspected Male genital tuberculosis, we usually look for M. tuberculosis in the urine or tissue. Traditionally, the appearance of so-called sterile pyuria on microscopic urinalysis is considered to be a typical manifestation of urogenital involvement. Some finds reported that leukocytes in urine were present microscopically or grossly in a majority of cases (50% and 10%, respectively). Hematuria is a common symptom of urinary tuberculosis, which is mainly caused by renal tuberculosis and bladder tuberculosis Hematuria and acidic urine have been associated with urinary tuberculosis, but they are nonspecific findings. In our patient cohort 17.0% were positive for red blood cells in urine, which is related to renal tuberculosis. Leukocyte positivity in the urine contributed to the diagnosis in 20 (42.6%) patients, but its specificity was low. Color Doppler ultrasound is the first choice for imaging analysis of epididymal tuberculosis (Viswaroop, Kekre & Gopalakrishnan, 2005). CT and MR have little value in diagnosing epididymal tuberculosis; they are mainly used to diagnose tuberculosis in lung and kidney and provide support for the diagnosis of epididymal tuberculosis. Epididymal fine needle biopsy is a good method for the diagnosis of epididymal tuberculosis.

Tuberculous epididymitis can be the only manifestation of genitourinary tuberculosis. Therefore, even in the absence of clinical and laboratory markers of renal and urologic tuberculosis, all men with identifed epididymal lesions should undergo a fine needle aspiration biopsy. Kim et al. (1993) suggested that epididymal tuberculosis can often be diagnosed by B-ultrasound biopsy, which supports the above conclusions. Polymerase chain reaction has been an important method for diagnose. Combined with pathological biopsy, it can improve the diagnosis rating epididymal tuberculosis in recent years (Chawla et al., 2012). It has the characteristics of high sensitivity, high specificity and short turnaround tie. However, at present, the diagnosis of simple epididymal tuberculosis is difficult and there is no preoperative diagnostic method with high sensitivity and specificity. The clinical diagnosis of scrotal abscess and ulceration is not difficult. It is often possible to make a clear diagnosis by examining for acid-fast bacilli in samples of the ruptured tissue through pus or secretion smears.

The differential diagnosis of epididymal tuberculosis includes bacterial epididymitis, epididymal sperm granuloma, epididymal tumor and other diseases. Of the 47 patients, 41 had typical symptoms, signs, imaging findings or a history of tuberculosis and they could be diagnosed with epididymal tuberculosis. Three patients had no typical symptoms of tuberculosis and were misdiagnosed as epididymal masses because of the characteristics of the tumor (30–50 years, weight loss, solid epididymal mass). Three patients without the above symptoms required surgery because of the mass affecting life. Patients with bacterial epididymitis often have testicular-epididymis pain and scrotal swelling and heat (Banyra & Shulyak, 2012). Features include a shorter course of illness and symptoms are generally relieved after antibiotic treatment. Carl & Stark (1997) reported that epididymal tuberculosis should be highly suspected when patients have persistent or repeated epididymitis episodes and symptoms cannot be controlled after adequate antibiotic treatment.

Epididymal sperm granuloma mainly occurs in the head of the epididymis, characterized by a smooth solid mass and antibiotics and anti-tuberculosis treatment are ineffective. Epididymal tumors are rare, accounting for about 0.9% of male reproductive system tumors. Most tumors occur during periods of sexual activity in men between 20 and 40 years of age. The disease is characterized by slow growth, large volume, and no tenderness. CT helps to confirm the diagnosis.

Epididymal tuberculosis, like other tuberculosis diseases, requires early, regular, full-course, moderate, combined anti-tuberculosis treatment. The drug treatment method uses three to four anti-tuberculosis drugs for 6–9 months (Alsultan & Peloquin, 2014). Surgical treatment is necessary if there is no response to drug treatment or in cases of abscess formation. Because the early symptoms of epididymal tuberculosis are not obvious, abscesses or involvement of surrounding tissues such as the testicles have often developed at the time of treatment, so most patients need surgery. When there is active tuberculosis, anti-tuberculosis treatment must be performed before surgical treatment. In this study, Chest imaging of eight patients (17.0%) suggested that pulmonary tuberculosis had developed calcification, which proved that they were old tuberculosis. Forty-one (87.2%) were identified as epididymal tuberculosis and were given antituberculous therapy for 2–4 weeks before surgery. In 28 cases in our patient cohort (59.6%) presenting as a testicular painless mass, the epididymal-testis border was unclear, or sinus formation was found, suggesting that the surrounding tissues were invaded. Although surgical treatment is effective, patients still need regular anti-tuberculosis chemotherapy for 3–6 months after surgery and close follow-up.

Conclusion

Epididymal tuberculosis is easy to diagnose in patients with a history of tuberculosis. However, in isolated epididymal tuberculosis, early symptoms are not obvious and cases are typically advanced at the time of diagnosis. There are obvious symptoms and signs such as epididymal enlargement, falling pain and bead-like changes. Epididymal tuberculosis has invaded surrounding tissue by the time it is discovered. Surgical treatment combined with preoperative and postoperative chemotherapy is an effective treatment approach. When epididymal tuberculosis manifests bead-like changes, testicular involvement, or sinus formation, good results can be achieved with definitive epididymal-testicular surgery. This experiment involved a relatively small number of cases and the patients did not undergo needle biopsy. Therefore, further research is needed to further characterize this disease and refine the treatment.

Supplemental Information

Supplemental Information 1 Age, clinical signs and symptoms, diagnostic methods, and treatment effects were collected from the patients’ electronic medical records.

Click here for additional data file.

We thank professors Junsheng Bao, Lingjun Zuo, Gongjin Wu, Zizhen Hou, Ganping Zhong, Zhongjin Yue, Jiaji Wang, Panfeng Shang and Jianmin Duan for their help in research designed and the provision of case datas.

Additional Information and Declarations

Competing Interests

Author Contributions

Human Ethics

Data Availability

The authors declare that they have no competing interests.

Jiangwei Man conceived and designed the experiments, performed the experiments, prepared figures and/or tables, authored or reviewed drafts of the paper, approved the final draft.

Lei Cao performed the experiments, prepared figures and/or tables, authored or reviewed drafts of the paper, approved the final draft.

Zhilong Dong analyzed the data, prepared figures and/or tables, approved the final draft.

Junqiang Tian analyzed the data, authored or reviewed drafts of the paper, approved the final draft.

Zhiping Wang analyzed the data, contributed reagents/materials/analysis tools, prepared figures and/or tables, authored or reviewed drafts of the paper, approved the final draft.

Li Yang conceived and designed the experiments, authored or reviewed drafts of the paper, approved the final draft.

The following information was supplied relating to ethical approvals (i.e., approving body and any reference numbers):

The Ethics Committee of Lanzhou University Second Hospital granted approval to carry out the study within its facilities (Number: 2019A-079).

The following information was supplied regarding data availability:

The raw data is available in Data S1.

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
