# Peer review of "Diagnosis and treatment of epididymal tuberculosis: a review of 47 cases"

_PeerJ, doi:10.7717/peerj.8291_

## Round 0.1 · original submission · Major Revisions

I read your paper carefully and noticed that it was mainly a review of case reports. This si very far from what I believe PeerJ should accept (especially since there are several other Journals where this type of reports would be welcome. However, I wish to give you the opportunity to improve your paper and to bring it to a level where more than case reports woud be presented and discussed (partly in line with the suggestions of one of the reviewers). If you are willing to do this work,we may consider your submission again. If you do do, please, submit a detailed rebuttal. Also, be aware that your new revised submission may undergo anew round of review by the same o by different reviewers. I cannot, therefore, make any commitment about a final acceptance of your submission.

·

Basic reporting

The article entitled "Diagnosis and treatment of epididymal tuberculosis:A review of 47 cases" is interesting and usefull.
English should be inproved.
List of references is sufficient.
Information about patients is incomplete. Authors missed some important data. I have done my comments and recommendations in the text of this article.
The quality of figures is poor.

Experimental design

This study is a simple retrospective non-comparative. Authors analysed 47 cases of histologically-confirmed epididymal tuberculosis in patients treated in the hospital for four years - and it is rich experience. They have got their purpose - but result ic incomlete, as some important data were missed.

Validity of the findings

The study was descriptive, and in Conclusion there are no clear specific recommendations, common well-known words only. The data were good - but analysis and conclusion poor.

Additional comments

I support this study and wish success to authors, but this article need major revision. Sure, this may be brilliant.

·

Basic reporting

The Manuscript is clear and unambiguous, professional English used throughout except little mistakes
Literature references, sufficient field background/context provided except sourse marked in RED
Professional article structure, figures, tables. Raw data shared
Self-contained with relevant results to hypotheses.

Experimental design

Well done.
No comment

Validity of the findings

No comment

Additional comments

Well done interesting work. However please correct the text accordind changes marked RED. Please, add reference Banyra O, and Shulyak A. Acute epididymo-orchitis: staging and treatment. Cent European J Urol. 2012; 65(3): 139–143

Reviewer 3 ·

Basic reporting

no comment

Experimental design

no comment

Validity of the findings

no comment

Additional comments

This peice of Journal has education purposed, therefore it's agreed upon to be published.

---

## Round 0.2 · accepted · Accept

All three reviewers were happy with your revisions and supported the publication of your study. Congratulations.

·

Basic reporting

No comments

Experimental design

No comments

Validity of the findings

No comments

Additional comments

Congratulation, now it is perfect!

·

Basic reporting

no comment'

Experimental design

no comment'

Validity of the findings

no comment'

Additional comments

WELL DONE!
CONGRATULATIONS

Reviewer 3 ·

Basic reporting

Clear and unambiguous, professional English used throughout. Also have enough reference.

Experimental design

no comment

Validity of the findings

Conclusions are cleared and acceptable.

Additional comments

This piece of journal has education purposes, therefore it is agreed upon to be published